# Unbalanced Sphingolipid Metabolism and Its Implications for the Pathogenesis of Psoriasis

**DOI:** 10.3390/molecules25051130

**Published:** 2020-03-03

**Authors:** Katarzyna Bocheńska, Magdalena Gabig-Cimińska

**Affiliations:** 1Department of Medical Biology and Genetics, University of Gdańsk, Wita Stwosza 59, 80–308 Gdańsk, Poland; katarzyna.bochenska@phdstud.ug.edu.pl; 2Institute of Biochemistry and Biophysics, Polish Academy of Sciences, Laboratory of Molecular Biology, Kładki 24, 80–822 Gdańsk, Poland

**Keywords:** sphingolipid metabolism alterations, skin barrier disruption, pathogenesis of psoriasis, lysosomal dysfunction

## Abstract

Sphingolipids (SLs), which have structural and biological responsibilities in the human epidermis, are importantly involved in the maintenance of the skin barrier and regulate cellular processes, such as the proliferation, differentiation and apoptosis of keratinocytes (KCs). As many dermatologic diseases, including psoriasis (PsO), intricately characterized by perturbations in these cellular processes, are associated with altered composition and unbalanced metabolism of epidermal SLs, more education to precisely determine the role of SLs, especially in the pathogenesis of skin disorders, is needed. PsO is caused by a complex interplay between skin barrier disruption, immune dysregulation, host genetics and environmental triggers. The contribution of particular cellular compartments and organelles in SL metabolism, a process related to dysfunction of lysosomes in PsO, seems to have a significant impact on lysosomal signalling linked to a modulation of the immune-mediated inflammation accompanying this dermatosis and is not fully understood. It is also worth noting that a prominent skin disorder, such as PsO, has diminished levels of the main epidermal SL ceramide (Cer), reflecting altered SL metabolism, that may contribute not only to pathogenesis but also to disease severity and/or progression. This review provides a brief synopsis of the implications of SLs in PsO, aims to elucidate the roles of these molecules in complex cellular processes deregulated in diseased skin tissue and highlights the need for increased research in the field. The significance of SLs as structural and signalling molecules and their actions in inflammation, in which these components are factors responsible for vascular endothelium abnormalities in the development of PsO, are discussed.

## 1. The Contribution of Selected Cellular Organelles In Lipid Metabolism

Sphingolipids (SLs) occur in the plasma membranes of all eukaryotic cells. The common element of SLs is the unsaturated 18-carbon amino alcohol sphingosine (Sph) or one of its analogues. The fatty acid in SLs is connected by an amide bond to the amino group of Sph. The combination of fatty acid with sphingosine gives the basic building element of all SLs—ceramide (Cer) [1]. Cer is the centre of SL metabolism. It is a precursor to all other types of SLs. The formation of Cer can occur via three different pathways in the appropriate cellular organelles (Figure 1) [2]:**de novo synthesis pathway on the cytoplasmic side of the endoplasmatic reticulum**. The first stage is the synthesis of 3-ketosphinganine, which consists of the condensation of L-serine and palmityl-CoA catalysed by the enzyme serine palmitoyltransferase. It is a key enzyme in the regulation of the synthesis of 3-ketosphinganine. In the next stage, this compound is reduced to sphinganine, which is acylated to dihydroceramide by the corresponding synthases. The last stage of the synthesis is the oxidation of dihydroceramide to ceramide catalysed by dihydroceramide desaturase [2,3,4].**salvage pathway in lysosomes**. Complex glycosphingolipids, which are important for the construction of biological membranes, undergo cyclic catabolism. Released Cer, with the participation of acid ceramidase, is converted into Sph. Studies show that Sph leaves the lysosome and can be metabolised to Cer or phosphorylated to sphingosine-1-phosphate (S1P). Kitatani et al. indicate that this pathway is responsible for 50–90% of SL biosynthesis [5].**catabolic pathway on the cellular membrane.** As a result of hydrolysis of sphingomyelin (SM) in the presence of the corresponding enzymes, sphingomyelinases (SMases), Cer and phosphocholine are formed. External factors, such as treatment with tumour necrosis factor α (TNF-α) or oxidative stress [5], cause hydrolysis of SM due to activation of sphingomyelinase. The classification of this enzyme is based on the differences in optimum pH values of the catalysed reaction and subcellular distribution; therefore, it stands out as acid SMase (aSMase; encoded by *ASAH1*); neutral SMases (N-SMases; encoded by *ASAH2*, *ASAH2B* and *ASAH2C*) and alkaline SMase (alk-SMase; encoded by *ENPP7*). Acid sphingomyelinase is located inside the cell and has the highest activity at pH 4.0–4.5, while the neutral one is placed on the outside of the plasma membrane and has optimal activity at pH 7.4. Moreover, it requires the presence of magnesium (Mg^2+^) or manganese (Mn^2+^) ions as an activator. SM degradation through activity of acid and neutral sphingomyelinase lasts a few seconds compared to de novo synthesis of Cer, whose duration is at least several hours [5,6].

Newly synthesized Cer can be further converted into five different lipids: ceramide-1-phosphate (C1P), sphingomyelin (SM), ceramide phosphoethanolamine (CPE), galactosylceramide (Galβ1–1′Cer) and glucosylceramide (GlcCer) [7,8].

Further changes towards more complex lipids occur in the Golgi apparatus. Transferases, located on the cytoplasmic side of the cisterns, attach the sugar residue to the Cer, forming cerebroside. In this shape, it is transferred inside the Golgi apparatus to extend the carbohydrate chain by attaching subsequent residues. Synthesis of sphingomyelin, which is based on Cer, occurs solely within the cisternae in the *cis* part of the Golgi apparatus [9,10]. For further transformation into glycosphingolipid, Cer is transported to the Golgi apparatus as an integral part of the membrane-bound transport vesicles of the endoplasmic reticulum (ER). For the production of sphingomyelin, a specific transporting protein (ceramide transfer protein (CERT)) is necessary to deliver Cer. For both SM and glycosphingolipids (GSLs), the final destination is the outer layer of the cell membrane [11,12].

However, the significance of two SL derivatives, C1P and S1P, in the biochemistry of each eukaryotic cell should be noted. Despite S1P not containing fatty acids, it is not included in the SL family, but due to its functional and metabolic similarity, it is often discussed with them [13]. S1P synthesis requires a sphingosine kinase (SphK), an enzyme, which phosphorylates sphingosine, while C1P is formed by ceramide kinase (CERK), adding orthophosphate to the Cer core [13,14,15]. It is highly important that both molecules have antagonistic effects; therefore, they should be considered as significant metabolites of the SL pathway [16].

The intracellular level of S1P is a result of a balance between its synthesis by SphK and its degradation catalysed by S1P phosphatase (SPP) and S1P lyase (S1PL) [17]. SphK catalyses phosphorylation of sphingosine to S1P and, thus, establishes balance between S1P and sphingosine. Two isoforms of this enzyme are characterized: sphingosine kinase 1 (SphK1) and sphingosine 2 kinase (SphK2). SphK2, unlike Sphk1, is found mainly in the cytosol, depending on the cell type, and is present in different intracellular compartments. Both kinases can move to the cell membrane, where they catalyse the synthesis of the pool of extracellular S1P [18].

## 2. Sphingolipids As A Structural Molecules - Role In Skin Barrier

Skin, the soft outer tissue of vertebrates, is a sort of “jacket” with three main functions: protection, regulation and sensation. In humans, it covers the body, guarding the underlying muscles, bones, ligaments and internal organs. The skin is known as the largest organ of the integumentary system. It consists of three main layers with an origin from up to seven coats of ectodermal tissue, resulting in a skin barrier composed of a mechanical barrier, a permeability barrier and innate and adaptive immunity barriers [19].

Human skin acts as a first line of defence, protecting the body from unwanted environmental influences. Although, we may be prone to regard the integument as a barrier against a hostile environment, it should be remembered that the most important task for human skin is to create a watertight enclosure of the body to prevent water loss. In fact, the development of such a permeable barrier was an essential step in the evolution of life on dry land. The actual barrier is located in the outermost layer (i.e., enucleated cells of the lipid-based stratum corneum called corneocytes), since once this part of the skin has been removed, substances are allowed to diffuse easily into or out of the body. One may place this fact into perspective by noting that man has a large surface area relative to the volume enclosed by the integument. Underlying the stratum corneum is the viable epidermis (i.e., a dynamic, constantly self-renewing tissue), in which a loss of the cells from the surface of the stratum corneum is balanced by cell growth in the lower epidermis (i.e., cell proliferation), followed by their maturation (i.e., cell changes in both structure, composition, synthesis and expression of numerous different structural proteins and lipids) and differentiation (i.e., cell transformation into corneocytes) [20]. The densely packed corneocyte envelope (cellular) is chemically linked to the lipid envelope (intracellular) of the stratum corneum, which serves as an interface between the hydrophilic corneocytes and the lipophilic extracellular nonpolar lipids surrounding the interconnected corneocytes. This structure is not only a physical barrier but, also, a space for many biological and biochemical reactions [21]. To provide a competent skin barrier, the lipids in the stratum corneum are highly specialized. There are virtually no phospholipids in this zone, whereas the major constituents are ceramides (belonging to a major class of lipids—sphingolipids, SLs); free fatty acids (mostly saturated and unbranched, with the predominant two species: lignoceric acid and cerotic acid) and cholesterol (belonging to a major sterol in the skin lipid barrier) in an approximately 1:1:1 molar ratio.

The proper function of the skin barrier requires many interactions between genetic and immunological factors that control the expression of proteins and enzymes controlling the lipid metabolism necessary for appropriate skin formation, composition and organization [22]. An important role in these processes and phenomena, resulting in a properly functioning skin barrier, is attributed to the correct metabolism of a prominent group of lipids—SLs. In the human skin, the presence of unusual SLs based on ultra-long fatty acyl chains as well as fatty acyl groups linked on the ω-end of the N-linked acids (thereby generating a three-chain rather than a two-chain molecule) were found. The sphingoid bases of SLs are enzymatically modified to generate a wide range of compounds, including ceramides, S1P, ceramide-1-phosphate (C1P) and glycosphingolipids (GSLs) [21].

Ceramides, namely N-acylsphingosines, as part of extracellular lipids in the stratum corneum, representing approximately 50% of the stratum corneum’s lipid content by weight, are important elements of the skin barrier and are involved in the prevention of transepidermal water loss. In the human body, ceramides are equally important inside the cell, because they regulate several cellular processes, such as proliferation, differentiation and apoptosis [22,23]. The basic structure of this main epidermal SL is a sphingoid base with a fatty acid connected by an amide bond. Four types of fatty acids with four types of sphingoid bases create 16 classes of Cers, being distinguished among themselves by the length of both of these moieties. The Cers of the skin can contain sphingosine (S) and dihydrosphingosine (dS), which are common in eukaryotic cells; phytosphingosine (P), which is found only in some human tissues and 6-hydroxysphingosine (H), which is specific for the epidermis [24].

Another important SL, S1P, is a phosphosphingolipid that consists of sphingosine with a phosphate group attached at position 1 [25]. It has attracted particular attention for its effect on epidermal cells, which differs from those on most other cell types. S1P inhibits proliferation of keratinocytes (KCs) (the most abundant cells in the epidermis) and programmed cell death (apoptosis), while it induces the differentiation and migration of KCs, suggesting a role for S1P in the re-epithelialization of wounds [26].

The skin lipids are synthesized in KCs and converted to their more polar precursors (i.e., Cers are converted into sphingomyelins and glucosylceramides, while S1P is converted into sphingosine) [26], which is stored in lamellar granules together with catabolic enzymes. The lamellar granules migrate from the lower to the upper surface of the KC, merge with the plasma membrane and secrete their contents into the intercellular space. The enzymes are activated and convert lipid precursors to barrier lipids that finally assemble into lamellar structures that occupy the entire space between the cells [27].

Diseased skin is often characterised by reduced barrier function and altered SL formation, composition and metabolism. Disruption of skin equipped with physical, chemical and immunological barriers leads to an altered response of intricate cellular pathways. Loss of function of several enzymes involved in SL metabolism leads to various forms of skin and hair defects that are often fatal [28].

### Skin Barrier Dysfunction in Psoriasis

Long-term studies of the metabolism of sphingolipids (SLs) linked to the formation of the human skin barrier have provided evidence of their role in the pathogenesis of numerous dermatologic diseases. A very interesting fact reported in the literature was the observation of the dysregulation of SL metabolism; in particular, quantitative and qualitative alterations of Cers and sphingosine, being a central hub of the SL pathway, in inflammatory skin diseases, such as psoriasis (PsO) and atopic dermatitis [28,29,30,31]. PsO, considered to develop through the crosstalk between epidermal keratinocytes and immunocytes, is a polygenic, chronic, inflammatory skin disease affecting about 2%–3% of the world’s population [32]. It is characterised by keratinocyte hyperproliferation, abnormal epidermal differentiation, and infiltration of immune cells into lesions. Many studies have reported various factors contributing to the pathogenesis of psoriasis, including genetic factors (mainly genes of HLA class I); the immune system (dendritic cell subpopulations, monocytes/macrophages, neutrophils and T cells, especially Th1, Th17 and Th22) and environmental conditions (such as mental stress, physical injury, infections and medications), thus recognizing it as a multifactorial disease [33]. Some reports have led to a change in the classification of psoriasis from “skin disease” to “T lymphocyte-mediated disease” [34]. Based on recent studies, it is worth noting that the majority of the pathogenesis of PsO is in fact concentrated on the skin barrier, which is confirmed by the study of a single nucleotide polymorphisms (SNPs) in the genes coding for the cornified envelope as well as enzymatic catalysis process proteins and the observations of changes in quantitative and qualitative disturbances of the extracellular matrix lipids.

Reports on the pathogenesis of psoriasis presenting interesting results obtained by genome-wide association study (GWAS) has included the genetic determination of the skin barrier dysfunction. Researchers’ attention has been focused on identification of novel genetic markers associated with the functioning of the skin barrier in PsO. In recent times, ten epidermal genes have been documented as psoriasis susceptibility genes [35]. As mentioned above, the role of the cornified envelope is crucial for proper functioning of the epidermal barrier. Therefore, any abnormalities in the expression of genes coding for proteins, which are a part of the envelope structure or participate in the enzymatic catalysis process, may result in disturbances at various phases of keratinocyte differentiation and ultimately lead to the dysfunction of the epidermal barrier [36]. It is assumed that the decrease in the synthesis of Cer and the malfunction of enzymes essential for its proper formation, being accompanied by a reduction in the apoptotic signalling molecules protein kinase C-alpha (PKC-α) and c-jun N-terminal kinase (JNK) [37], may be correlated with the severity of psoriasis [38]. Reports on analyses of patients with PsO have shown a significant decrease in the synthesis of Cer in skin lesions [39], while others proved that the qualitative changes in the composition of Cer species in the epidermis [40]; in both cases, they affect the epidermal barrier. What is more, two other cases revealing the modulation in Cer formation processes were described in the literature [41,42,43]. At the same time, the observations of changes in quantitative and qualitative disturbances of the extracellular matrix lipids have showed that the level of sphingosine is markedly higher in the psoriatic epidermis compared to skin without lesions [38,41,42]. Yet another aspect that is worthy of attention is the role of antimicrobial peptides and proteins (AMPs) as barrier components of innate immunity defence. AMPs, of which more than 20 are known to be present in human skin, can recruit leukocytes to the skin and stimulate them to release cytokines and chemokines [35,43]. In patients with psoriatic lesions, there are three subclasses of AMPs, including cathelicidin (LL37), S100 proteins (S100A7/8/9) and defensins (hBD2/3), which are highly expressed and considered to play an important role in the pathogenesis of psoriasis [44]. Concluding the topic of the skin barrier dysfunction in PsO, it is worth noting that current thinking supposes that psoriasis develops as a result of the vicious cycle of barrier defect, innate immunity and adoptive defence, each step of which may be linked genetic predisposition. In detail, this cycle comprises: (i) barrier insults by trauma, infection and other mechanisms, (ii) abnormal response of barrier recovery, which may be due to intracellular signalling or lipid deficiency, (iii) resulting in enormous AMPs and abnormalities of epidermal proliferation and differentiation and (iv) all of which may lead to excessive, uncontrolled immune aberration towards Th 17/23 [35].

## 3. Sphingolipids As Signaling Molecules

The prefix sphingo- was used to express properties similar to the sphinx of this mysterious class of lipids. The most important function assigned to SLs is that they act as second messengers in many biological processes [45].

### 3.1. Sphingolipid Role in Cellular Processes – Deregulation Of Keratinocytes’ Function in Psoriasis

In mammalian skin, SLs are not only an essential component of cell membranes but also are involved in the significant biological processes, such as the aging, differentiation, growth regulation and apoptosis of epidermal cells. During these processes, SLs metabolism is continuously changing [26].

#### 3.1.1. Cell Death In Skin Homeostasis

In the skin, apoptosis (i.e., programmed cell death) plays a key role in maintaining epidermal homeostasis through the regulation of proliferation and formation of keratinocytes in the stratum corneum. An important step of the stratum corneum formation is cornification, which is reported as the conversion of living keratinocytes into dead corneocytes. Cornification is a specific variety of apoptosis, during which the typical content of the cell is replaced by a cytoskeleton. Crosswise interconnected proteins around the periphery of the keratinocytes form a horny envelope, while modified and tightly packed lipids fill the intercellular spaces, forming a barrier [46]. The cornification process is impaired in psoriasis as evidenced by late cornified envelope (LCE) gene cluster research [47]. Studies on programmed cell death in psoriasis showed a decreased apoptosis index in psoriatic skin lesions in comparison to normal epidermis, which can be caused by high expression of epidermal growth factor (EGFR) and interleukin-15 (IL-15RA) receptors on keratinocytes [48]. Many studies indicate the important role of SLs in the regulation of apoptosis, especially ceramide (Cer), which seems to be an important molecule in regulating this process [49,50].

Programmed cell death is an active process requiring the triggering of many molecular reactions and energy expenditure. As a result of exposure of cells to stress or viral infections, a sphingomyelin-Cer pathway may be activated. This pathway is associated with an increased concentration of Cer in the cell, which results from the connection of a suitable ligand with a tumour necrosis factor family receptor (TNFR), i.e., Fas ligand (FasL) or IL-1, followed by activation of aSMase or NSMase. This enzyme cuts sphingomyelin to form Cer and phosphocholine [49]. Cer acts as a secondary lipid transmitter that can activate specific protein targets, such as protein kinase B (Akt), protein kinase C (PKC), mitogen-activated protein kinase (MAPK), kinase cascades stress associated protein kinase/Jun N-terminal kinase (SAPK/JNK), Cer-activated protein phosphatase (CAPP) and phospholipase D (PLD) [50]. Cer can also activate other cellular processes and affects the activation of the internal pathway of apoptosis by increasing the permeability of the mitochondrial membrane and release of cytochrome c [51].

In contrast to Cer, sphingosine-1-phosphate (S1P) plays the opposite, prosurvival role by enhancement of cell inflammation, proliferation and simultaneously resistance to apoptotic signals. The anti-apoptotic effect of S1P is associated with several cellular processes [52]. In some cell populations, S1P may prevent apoptosis through inhibition of changes in the membrane potential of mitochondria, thus preventing release of cytochrome c. In other cells, S1P directly stimulates MAPK, leading to the activation of Akt kinase and inhibition of several caspase family proteins [53].

It is interesting to note that sphingosine kinase 1 (SphK1) and sphingosine 2 kinase (SphK2) have different effects on cell survival. During studies on various cell types, SphK1 hyperactivity and increased S1P synthesis were found to promote cell survival by inducing their transition from the G1 phase to the S phase. Compared to SphK1, the hyperactivity of SphK2 inhibits cell growth and enhances apoptosis. In vitro studies on fibroblasts showed that overexpression of the gene encoding SphK2 results in cell cycle arrest, activation of caspase 3 and release of cytochrome c, which ultimately leads to apoptosis [54]. It is believed that S1P formed with the participation of SphK1 can act directly or indirectly as a negative regulator of enzymes involved in the synthesis of Cer, such as palmitoyltransferase serine (SPT 1) or ceramide synthase (CerS). This regulation of Cer biosynthesis by S1P can lead to inhibition of apoptosis dependent from Cer. In contrast, SPP phosphatase may also indirectly regulate the concentration of Cer through reduction of the concentration of S1P, consequently inducing the process of apoptosis [55,56].

Moreover, research done by Bonhoure et al. indicated that the overexpression of SK1 (encoding sphingosine kinase 1) inhibits caspase 3 activation and cytochrome c release through B-cell lymphoma-extra-large (BCL-XL), induced myeloid leukaemia cell differentiation protein (Mcl-1) and Bcl-2-like protein 11 (BIM)-dependent pathways and leads to increases in the cell proliferation rate by G1/S phase transition and increases in DNA synthesis [57]. Exogenous S1P may prevent the intrinsic apoptotic pathway by blocking the transition of BCL2 associated agonist of cell death/apoptosis regulator BAX (BAD/BAX) to mitochondria. However, knockdown of SK1 results in inhibition of cell proliferation [58,59,60,61,62,63,64].

Acid (ACDase) and alkaline (AlkCDase) ceramidase intensify Ca^2+^-induced cell cycle arrest and differentiation of keratinocytes. Overexpression of AlkCDase 2 increases β1 integrin maturation and cell adhesion. AlkCDase 2 and 3 have been shown to especially coordinate keratinocyte proliferation and apoptosis [65].

#### 3.1.2. Cell Proliferation

Maintaining the homeostasis of cells during their development is possible due to the precisely regulated balance between cellular proliferation and death. The human epidermis consists of approximately 97% keratinocytes. Within 19 days, they pass from the basal layer to the stratum corneum accompanied by morphological changes based on their biochemical level. The basal layer contains stem cells, which retain their constant number during division and give rise to a population of transiently multiplying cells. These have high proliferative potential only for a limited time, after which the cells differentiate. During this process, they undergo many structural changes, including proteomic modulation along with the reduction of the total cell biomass. Homeostasis of the epidermis is regulated by many endogenous and exogenous factors (i.e., Ca^2+^ concentration, cytokines and UV irradiation), affecting the formation of the skin barrier. Moreover, the architecture of the epidermis is largely regulated by the action of lysosomes activating signalling pathways and are responsible for the intracellular digestion necessary for the proper differentiation of keratinocytes [66,67,68].

In psoriatic keratinocytes, the time of cell division is shortened from 457 h to 37.5 h. Research by Pasquali et al. revealed that, in the transcriptome of keratinocyte from psoriasis lesions, many genes associated with the cell cycle are modulated [69]. The hyperproliferation observed in psoriasis results from reduced apoptosis, lack of maturation of keratinocytes and incomplete keratinization with retention of nuclei in the stratum corneum (parakeratosis). Increased proliferation may be due also to the presence of epidermal growth factor receptor (EGFR) on the surface of keratinocytes of all layers of the epidermis and not only on basal layer cells, as in healthy skin [70]. Disability of catabolic processes, such as autophagy and nucleophagy, observed in keratinocytes with a psoriatic phenotype and impaired skin barrier, arising from the altered composition of Cers and SLs, may result from quantitative and qualitative disturbances of lysosomes and their enzymes, as we have shown in our research. As a result of gene expression studies, we observed modulation of LAMP1 encoding lysosomal-associated membrane protein 1 marker both in the “psoriasis-like” inflammation in vitro model (HaCaT cells) and in the majority of patients (both psoriatic plaque (PP) and psoriatic normal (PN) compared to nonpsoriatic normal skin (NN)) [71]. Moreover, vesicular organelles, such as lysosomes, seem to be the obvious candidates playing an important role in the inflammatory process of PsO, as evidenced by the gene expression and specific pathways analyses of our earlier studies [72]. In turn, our ongoing research shows significant changes in the quantity of lysosomes and other acidic organelles, as well as dysregulation of endosomal-lysosomal signalling both at the in vitro level (in keratinocytes with psoriatic phenotype) and at the in situ level (in the skin of psoriatic patients, both PP and PN compared to NN). Changes in the number of lysosomes may result in the dysregulation of selected enzymes (i.e., those involved in the Cer conversion pathway), leading to a lack of differentiation and excessive proliferation of keratinocytes, inhibition of apoptosis and disturbed skin barrier (data not published).

In the homeostatic state, S1P, by the activity SphK1, protects cells from Cer-induced apoptosis by stimulation of cell proliferation and motility, while the activity of SphK2 shifts the function of S1P to induce cell cycle arrest. In keratinocytes, S1P induces differentiation but does not affect proliferation of these cells [73]. An important mechanism by which extracellularly released S1P controls cellular metabolism is the activation of the plasma membrane localized G-protein coupled S1P receptor (S1PR). So far, five isoforms of S1P receptors (from S1PR1 to 5) have been characterised, and all of them are expressed in keratinocytes [74]. Many signalling pathways, such as phospholipase C (PLC), phosphoinositide 3-kinase (PI(3)K) and protein kinase B/ras-related C3 botulinum toxin substrate 1 (Akt/Rac1), protein kinase C delta type (PKCδ) and mothers against decapentaplegic homolog 3 (Smad 3), as well as release of Ca^2+^ ions, remain under control of S1P receptors [75]. Depending on the type of activated receptor, S1P can both increase and decrease cell proliferation. By acting via the S1P1 and S1P3 receptors, S1P can increase cell migration. However, when S1P activates the S1P2 receptor subtype, this process is inhibited (mainly by blocking the Akt signalling pathway, which is essential for keratinocyte proliferation). Cell proliferation is also influenced by growth factors interacting with S1P. Both S1P concentration and SphK activity modulate the cellular response to a number of factors, such as platelet growth factor (PDGF), epidermal growth factor (EGF) and transforming factor growth β (TGF-β). In addition to interacting with receptors, S1P also modulates cell activity through receptor independent pathways, such as the modulation of histone acetylation or by polyubiquitination of receptor interacting protein 1 (RIP1) and nuclear factor kappa-light-chain-enhancer of activated B cells (NF-κB) activation in the RelA pathway [76,77].

On the other hand, one of the interesting and quite well-described manifestations of Cer activity is inhibition of telomerase activity. Rebuilding telomeres is necessary for a cell to remain in the cell cycle and repeat divisions. Meanwhile, Cer inhibits the expression of telomerase coding genes by inactivating the transcription factor c-Myc. Thus, the cell is forced to stop divisions [78]. The balance between Cer, sphingosine and S1P concentrations determine cell fate (death or survival) (Figure 2). Its shift towards increased Cer and sphingosine concentrations leads to cell death. In turn, cell survival and/or proliferation requires high S1P concentration [79].

#### 3.1.3. Keratinocytes’ differentiation

The process of keratinization of epidermal cells is a genetically conditioned, multistage biological process involving cell division, differentiation and death. The skin barrier formed in this process not only prevents excessive water loss but also protects against the damaging effects of the adverse external environmental factors. The key signalling factor in the keratinocyte differentiation, which is essential for all stages, is the calcium ion — Ca^2+^. Both desmosome formation, lamellar body secretion and activation of enzymes involved in differentiation are calcium-dependent processes. Its concentration in the extracellular space varies in the individual layers of skin, and it is around four times higher in the upper parts of the epidermis (between 1.2 and 1.8 mM) compared to the basal layer (from 0.05 to 1.0 mM). The calcium gradient is sufficient to induce cell proliferation, but further transformation requires higher concentrations both in the extracellular space and in the cytosol [80,81,82]. The presence of the cell nuclei together with the increased proliferation rate of epidermal cells occurs in the course of psoriasis. The current state of knowledge indicates that psoriatic keratinocytes are characterised by lack of differentiation (decreased expression of K1, K10 and SIRT1 markers) compared to normal cells, which may result from impaired calcium homeostasis [83].

SLs also affect the level of calcium ions in the cell. Cer releases calcium from the ER and leads to an increase in calcium content in the cytoplasm and mitochondria. Research on normal cultured human keratinocytes showed that Cers are strongly involved in keratinocyte differentiation by activation of apoptosis signal-regulating kinase 1 (ASK1) and enhancing synthesis of caspase-14 [78,79].

S1P may also be involved in the regulation of calcium homeostasis in mammalian cells. In keratinocytes, increased intracellular levels of S1P not only regulate migration and growth of these cells but also increase Ca^2+^ concentration. Research suggests that S1P affects intracellular levels of Ca^2+^ through S1P2 and S1P3 receptors and releases Ca^2+^ ions mainly from the ER. S1P can also induce release of calcium ions from permeabilised cells by a non-receptor mechanism dependent on inositol triphosphate (IP_3_) [80]. The increase in the level of intracellular S1P directly mobilizes the Ca^2+^ ions from thapsigargin (TG)-sensitive stores and therefore increases cytosolic Ca^2+^ levels [84,85]. In turn, the increased quantity of extracellular S1P activates store-operated calcium entry (SOCE) through a mechanism dependent on G-protein coupled receptors [86,87]. As mentioned, Sphk1 is necessary for the synthesis of S1P and, thus, indirectly enhances gene expression of the differentiation markers, such as filaggrin (FLG) and involucrin (IVL), as studies on HaCaT (Ha = human adult, Ca = calcium and T = temperature) keratinocytes have shown. In contrast, blocking SPHK1 leads to apoptosis in UVB-treated keratinocytes [88].

### 3.2. Sphingolipid Action In Inflammation—Focus On The Molecular Mechanism Of Psoriasis

Particularly important for local immunity in the skin is the secretion of large amounts of antimicrobial peptides, mainly β-defensins (hBDs), cathelicidins (LL-37 and CAMP) and psoriasins (S100A7/8/9). In the early stages of psoriasis (PsO), plasmacytoid dendritic cells (pDCs) are activated by complexes of the antibacterial peptide cathelicidin LL-37 and self-DNA/RNA in a mechanism similar to activation of toll-like receptor 9 (TLR9), which may explain how the host DNA becomes a pro-inflammatory stimulus, breaking the immune tolerance in PsO. S1P, as an ER-stress signalling molecule, strongly stimulates the expression of CAMP, while the epidermis was exposed to stress factors (i.e., UV irradiation or attack by microorganisms) [89]. Knockdown of S1P lyase, (a polar SL metabolite) which catabolizes S1P, promoted ER stress-induced CAMP production in cultured keratinocytes and in mouse skin [90].

Complexes of LL37 and self-DNA released from keratinocytes activate pDCs to secrete interferon α/β (IFN-α/β). pDCs, also referred to as interferon-producing cells, belong to key cells linking the innate and acquired immune responses. In normal conditions, pDCs, unlike myeloid dendritic cells (mDCs), are hardly found in the tissues other than lymphatics [91]. However, in inflammation accompanying diseases, such as lupus erythematosus or psoriasis, infiltration of the pDCs in peripheral tissues is often observed. Within PsO lesions, they induce T cell autoproliferation and cytokine production, mainly by T helper cells (Th cells, CD4^+^ cells) [92,93].

The role of Cers in the activation of dendritic cells (DCs) is constantly expanding. The first studies were focused on an induction of DC apoptosis by C2-Cer with simultaneous downregulation of the survival signalling pathways, such as nuclear factor kappa-light-chain-enhancer of activated B cells (NF-κB), phosphoinositide 3-kinase (PI3K), protein kinase B (Akt) or B-cell lymphoma-extra-large (Bcl-xL). Another example is that the local administration of a cell permeable analogue of naturally occurring C8-Cer results in the induction of DC maturation, appearance of major histocompatibility complex (MHC) class I molecules and secretion of pro-inflammatory cytokines, such as IL-12p70 and TNF-α, in response to lymphocytic choriomeningitis virus clone 13 (LCMV Cl 13) and influenza virus infections [94]. On the other hand, S1P regulates the Th17-dependent inflammatory profile of DCs by induction of IL-6, IL-23 and STAT3, with simultaneous repression of Th1 responses by a reduction of IL-12p70 expression [95]. Moreover, application of fingolimod (FTY720), a structural analogue of sphingosine, to bone marrow-derived dendritic cells (BMDCs) induces alterations of surface markers, decline of shape indices and cell volume and surface roughness in lipopolysaccharide (LPS)-treated cells [96].

It has been demonstrated that SLs are necessary not only in the maturation and development of DC subtypes but also are equally important in modulating the activities of these cells. Data derived from research using the immunomodulating drug FTY720 suggest that it affects the reduction of pro-inflammatory cytokine production, including IL-6, IL-12, TNF-α and monocyte chemoattractant protein-1 (MCP-1) [96]. Moreover, S1P enhanced IL-6- and IL-23-dependent inflammation with reduction of IL-12p70, which suggests that using S1P as a medicine may be valuable for diseases indirectly mediated by DCs, such as psoriasis [95].

A crucial mechanism for the development of PsO is the interaction between dendritic cells and keratinocytes. S1P has anti-inflammatory effects by reducing the production of IL-12 and IL-23 in DCs via the common subunit p40, thus inhibiting the crosstalk of DCs with activated keratinocytes. Moreover, S1PR1 is highly important in the modulation of the cytokine profile via the mitogen-activated protein kinase (MAPK) pathway. Another significant intercellular communication occurs between DCs and naïve T cells. Dendritic cells migrate to regional lymph nodes, stimulating T cell activation and maturation in response to stimuli. This is an immunological synapse model in psoriasis. Recirculation of naïve T cells is dependent on S1P receptor signalling. In the first stage, the formation of T lymphocytes is dependent on a strong activation signal involving the T cell receptor (TCR) and co-stimulatory signal from the CD28 molecule, a process mediated by SMase activity. In turn, NSMase2 generates microdomains, regulating the signal relay for signalling proteins [97]. Activation of CD3 enhances NSMase2 production, while CD28 stimulates aSMase [98,99,100]. In the second stage of T cell maturation, cytokine signals are needed. In vitro studies showed that Cer inhibits the production of IL-2 (a cytokine that promotes naïve T cell differentiation into Th1 and Th2 cells) by blocking protein kinase Cθ (PKCθ), an effector molecule of CD28 signalling, mediating activation of NF-κB [101,102].

aSMase and NSMase are important enzymes involved in synapse formation in DCs and T cells. aSMase controls human CD4^+^ T cell activation by interaction with the intracellular domains of CD3 and CD28 membrane receptors and its downstream signalling components, including CD3-ZAP70-PLC-γ1- MAPK/JNK3 and CD28-PI3K-Akt-mTOR [103]. In contrast, pharmacological inhibition or knockdown of aSMase activity by small hairpin RNA (shRNA) blocks CD4^+^ T cell activation and proliferation. In addition to CD3/CD28 receptors interaction, aSMase mediates numerous signalling pathways. Pro-inflammatory cytokines (i.e., TNF) are capable of negating TCR signals and impair T cell activation and functions via its type 1 receptor (TNFR1). Exposure to TNF leads to an increase in the aSMase level with simultaneous inhibition of Ca^2+^ influx in Jurkat cells. Interestingly, blocking of Ca^2+^ responses by TNF/TNFR1 does not occur in aSMase-deficient murine T cells. These studies provide evidence that aSMase is strongly involved in TNF-dependent inhibition of Ca^2+^ and TCR signals in T lymphocytes. Moreover, CD39 and CD161 molecules interact with aSMase, thereby affecting downstream signals dependent on STAT3 and mTOR, which leads to polarisation of Th17 cells. Equally important, aSMase mediates Th1 responses but inhibits the Treg-dependent response [104,105].

The chronic stage of psoriasis occurs when mature dermal DCs and myeloid DCs produce IL-23 and IL-12 cytokines, which are necessary for the activation of Th1, Th17, Tc17 and Th22, which further produce pro-inflammatory agents acting on keratinocytes that contribute to the severity of the disease. Many studies have provided evidence of the role of individual SLs and their derivatives in the biology of T lymphocytes at various levels [106].

In addition, SLs are also relevant in T cell migration. Each of the five different S1P receptors plays a unique role in proliferation, trafficking and egress from the lymph nodes to the peripheral blood and in transport from the peripheral blood to target tissue. S1P increases the synthesis of adhesion molecules necessary for T cell recruitment and, together with S1PR4, are chemotactic for T cells. Studies conducted on psoriatic patients taking FTY720 indicated a reduced amount of central memory T cells and naïve T cells in the peripheral blood with simultaneous increasing amounts of T effector and Treg cells. S1P also inhibits 12-*O*-tetradecanoylphorbol 13-acetate (PMA)-induced proliferation of T cells in vitro [107,108].

Acid sphingomyelinases regulate pro-inflammatory activity of CD8^+^ T cells by regulation of cytotoxic granules and cytotoxic effector molecule secretion. In ASM-knockout CD8^+^ T cells, cytokine production, including IFN-γ, was decreased, indicating the key role of acid sphingomyelinases in controlling CD8^+^ T cell polarisation and functions [109]. In addition to CD8^+^ T lymphocytes, a large number of CD4^+^ T helper cells (mainly Th1 and Th17) were observed both in psoriatic skin lesions and the peripheral blood of patients. Moreover, increased levels of cytokines, including IFN-γ, TNF-α and IL-12, were also noticed, which defined PsO as a disease mediated by T helper cells. Besides Th1 lymphocytes, which were thought to play a major role in the pathogenesis of psoriasis, the Th17/IL-23 axis is also considered to be of great importance [110]. IL-23 is produced by DCs and other antigen presenting cells (APCs), which is required for the development and maturation of Th17 effector lymphocytes. Genetic studies have shown that IL-23p19, IL-12/23p40 and IL-23R polymorphisms are associated with and increased risk of developing PsO [111]. IL-23, together with transforming growth factor β (TGF-β), IL-1b, IL-6 and autocrine IL-21, is involved in the differentiation and maturation of Th17 lymphocytes, while TGF-β inhibits IL-22 production. Th17 cells affect the production of chemokines, antibacterial peptides and other pro-inflammatory cytokines, as well as the recruitment of neutrophils, facilitating a rapid development inflammatory reaction [112].

Continuous inflammation of psoriatic skin is caused by the excessive influx of immune cells, among which the dominant cells are neutrophils, and forms Munro’s microabscesses in the stratum corneum of the epidermis [113]. Cer also plays an important role in monitoring the function of neutrophils, particularly in response to TNF-α, through control of superoxide production, mainly by C2-Cer. C2-Cer blocks the respiratory burst of N-formylmethionine-leucyl-phenylalanine (fMLP)-stimulated adherent neutrophils. In addition, C16- and C24-Cers also play a role in the apoptosis mediated by caspase activation. In contrast, granulocyte macrophage colony-stimulating factor (GM-CSF) is an antagonist of this process, which reduces the accumulation of Cers in neutrophils and acts as a pro-survival factor [114].

In addition to the infiltration of neutrophils and DCs, macrophages and natural killer T (NKT) cells are also recruited into skin lesions (as well as skin without lesions, in smaller quantities) in PsO. Dermal macrophages, activated by cytokines released from T cells or DCs, produce large amounts of TNF-α, IL-12 and IL-23 in response to local S1P, leading to the skin changes observed in psoriasis. TNF-α signalling is orchestrated by intracellular S1P through targeting TNF receptor-associated factor 2 [115,116,117,118]. Moreover, S1P acts as a chemoattractant for monocytes, which further polarise into macrophages. Such differentiation induces these cells to secrete the proangiogenic factor prostaglandin E2. Macrophages express S1PR on their cell surface, mainly S1PR1 and S1PR2. M2 macrophages with anti-inflammatory phenotypes are induced by IL-4, whose secretion is mediated by S1P signalling. Additionally, S1PR1, but not S1PR2, downregulates inducible nitric oxide synthase (iNOS) signalling and enhances the formation of an anti-inflammatory phenotype in M2 macrophages [119]. In turn, S1PR2 together with S1PR3 are necessary for macrophage recruitment. S1P does not have as much impact on the phenotype of M1 cells [120].

### 3.3. Sphingolipids In Endothelial Function—Vascular Endothelium Abnormalities in the Development of Psoriasis

The vascular endothelium is a source of inflammatory mediators and surface particles that facilitate cell migration between the blood and tissues and, thus, takes part in the development of inflammation-related diseases. It has been observed that an increase in vascular endothelium permeability precedes the development of psoriasis and exacerbation symptoms [121]. In addition, vascular endothelium may actively support the development of a local Th2-type response. The location of endothelial cells makes them the first line of defence with regard to blood, and they have the first contact with cells and substances transported from the blood to tissues. This location allows a quick response to chemical and physical stimuli, which results in changes in the synthesis of adhesion molecules or the secretion of factors that regulate the proliferation and migration of vascular smooth muscle cells (VSMCs), tension in the walls of arteries and veins and the development of inflammation. Within PsO lesions, the process of angiogenesis is excessively stimulated, and numerous vascular abnormalities in histological and capillaroscopic examinations are found. Blood vessels of altered psoriatic skin are widened and elongated, and their permeability increases significantly. Patients suffering from PsO have an increased capillary mass and accelerated blood flow in skin lesions compared to normal tissue [122,123].

Cer and S1P have antagonistic effects on vascular endothelium (Figure 3). S1P is crucial for maintenance of vascular integrity and promotes angiogenesis directly by activating the vascular endothelial growth factor receptor (VEGFR) or indirectly by the release of vascular endothelial growth factor (VEGF) and interleukins (including IL-6 and IL-8) [115]. Beside VEGF, factors that stimulate sphingosine kinase include TNFα and EGF [116]. S1P also activates the type 1 extracellular matrix metalloproteinase (MT1-MMP), inducing endothelial cell migration [117]. Furthermore, the S1P-S1PR1 axis is involved in regulating the integrity of high endothelial venules (HEVs). Endothelial S1PR2 inhibits the PI3K-Akt pathway, suppressing nitrogen oxide (NO) generation, and protects against the destruction of adherent junctions [124]. In contrast, an increased Cer level (due to the activation of acid or neutral sphingomyelinase) leads to increased vascular permeability and endothelial barrier dysfunction [125,126]. Cer induces endothelial cell death by activation of caspases and increasing mitochondrial permeability, and they have also been linked to growth arrest, cytoskeleton rearrangements and, consequently, senescence of endothelial cells [127].

## 4. Conclusions

The discoveries made in recent decades have demonstrated the key role of SLs in cell signalling and cellular fate control through processes such as differentiation, proliferation and programmed cell death. In the skin, SL metabolites also act as bioactive molecules involved in the control of epidermal and immune cell signalling. Disorders of their metabolism underlie many pathological conditions, including dermatological and immunological diseases. Psoriasis (PsO) is a disease with a multifactorial nature, which means its exposure and severity are not only dependent on both a genetic predisposition and disturbances of the immune system but also on environmental impact. As a result of long-term overproduction of pro-inflammatory cytokines and the influx of T cells into the skin, there is excessive activation and overgrowth of keratinocytes, resulting in the formation of PsO plaques with painful and reddened changes. In the case of psoriasis, SLs act on many levels, which proves their multiple functions in the pathogenesis process, constituting the full picture of the disease. Their quantitative imbalance not only leads to skin barrier damage but also affect migration and maturation of T lymphocytes, activation of DCs or action of neutrophils, macrophages and natural killer T (NKT) cells. They are also an inherent factor for the proper structure and function of the vascular endothelium. The pathogenesis of psoriasis is a complicated process and still subject to the extension of current knowledge, where the development of lipidomics will contribute to understanding many aspects and mechanisms leading to the development of the disease. In addition to the fact that SLs perform multiple functions in the pathogenesis process in psoriasis, another level regarding the role of these compounds may be the use of their characteristics as a complementary or alternative way or remedy in the choice of treatment. In this light, for example, study of the efficacy and safety of ponesimod (a selective modulator of the S1P receptor 1) suggested that sphingolipid metabolism may become a new target for the pharmacological treatment of psoriasis [20]. It is also known that proper treatment of psoriasis is the best strategy for preventing it from spreading. Perhaps in the future this approach will prove to be even helpful in assessing the risk of this disease. The combination of all the fragmentary information on the role of SLs can provide much evidence to extend the full picture of PsO.

## Figures and Tables

**Figure 1 molecules-25-01130-f001:**
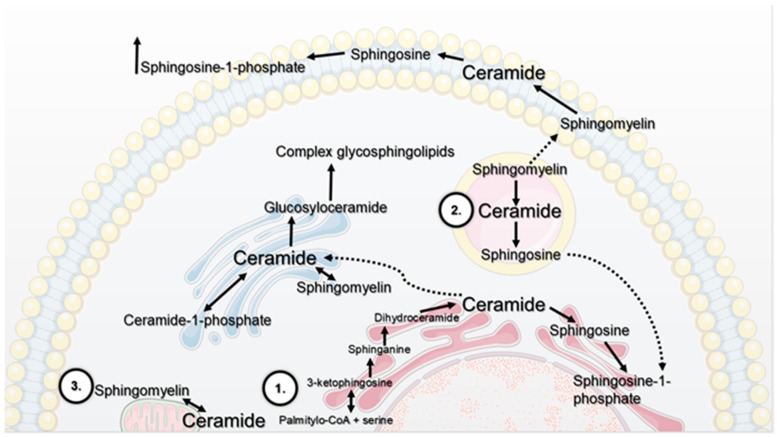
Key pathways of sphingolipid metabolism. De novo synthesis (1) of ceramide occurs mainly on the outer membrane. Ketosphinganine is formed as a result of the condensation of palmityl-CoA and serine. In the next stage, 3-ketosphinganine is rapidly converted to sphinganine. A fatty acid residue is attached to the sphinganine molecule, resulting in the formation of dihydroceramide. The final stage of de novo ceramide synthesis is desaturation, consisting in the formation of a double bond between carbons 4 and 5 in a dihydroceramide molecule. Then, ceramide is transported to the Golgi apparatus and metabolised to other sphingolipid groups. In lysosomes, the synthesis of ceramide from sphingosine under the influence of ceramide synthase is called the salvage pathway (2), where complex sphingolipids are digested to sphingosine, which can be transported to various cellular compartments. Ceramide may also be synthesized in the mitochondria-associated membranes (3).

**Figure 2 molecules-25-01130-f002:**
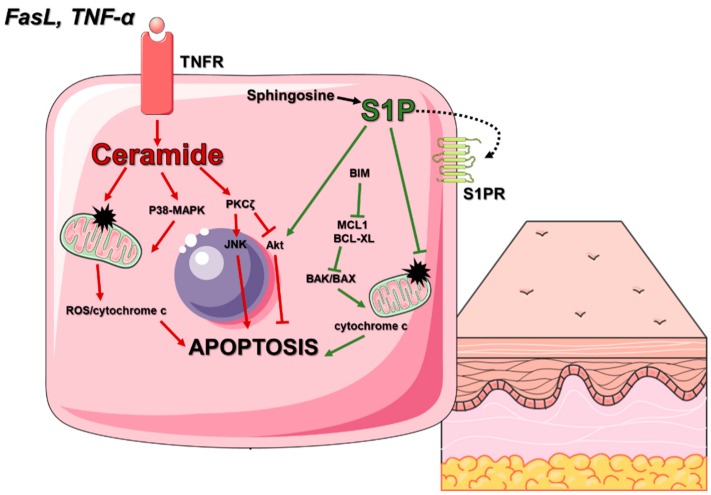
Role of the ceramide (Cer) and sphingosine-1-phosphate (S1P) in keratinocyte apoptosis and survival. Ceramide (activated by FasL or TNF-α) acts as a secondary lipid transmitter leads to apoptosis through stimulation of specific protein targets, such as P38-MAPK, PKCζ or JNK and inhibition of Akt kinase activity or through direct permeabilization of the mitochondrial membrane and release of cytochrome c into the cytoplasm. S1P acts as a ceramide antagonist, promoting cell survival by blocking a BIM-dependent signalling cascade, thus preventing the release of cytochrome c from mitochondria or by activating Akt. **Akt**—protein kinase B; **BAK**—Bcl-2 homologous antagonist killer; **BAX**—apoptosis regulator BAX; **Bcl-xL**—B-cell lymphoma-extra-large; **BIM**—Bcl-2-like protein 11; **FasL**—fas ligand; **JNK**—c-Jun N-terminal kinase; **MCL-1**—induced myeloid leukaemia cell differentiation protein; **P38-MAPK**—p38 mitogen-activated protein kinase; **PKCζ**—protein kinase C zeta type; **ROS**—reactive oxygen species; **S1P**—sphingosine-1-phosphate; **S1PR**—sphingosine-1-phosphate receptor; **TNF-α**—tumour necrosis factor alpha and **TNFR**—tumour necrosis factor receptor.

**Figure 3 molecules-25-01130-f003:**
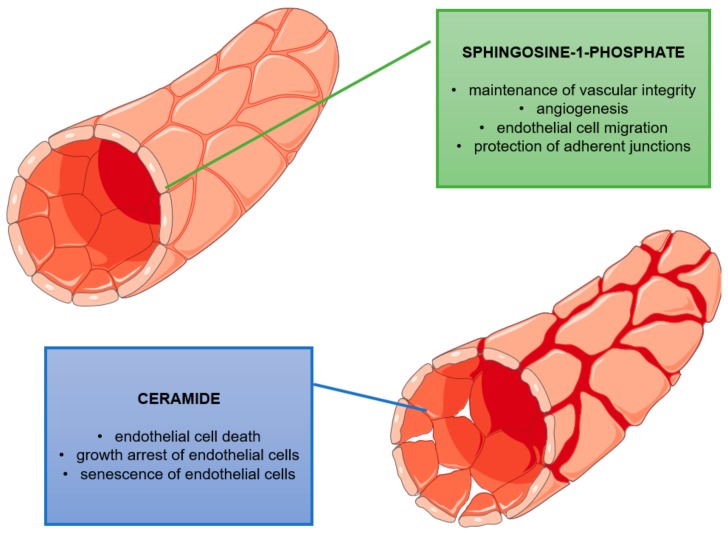
Role of ceramide (Cer) and sphingosine-1-phosphate (S1P) in the functions of vascular endothelium. S1P is necessary for maintenance of proper vascular integrity and promotion of angiogenesis. It is also a molecule that activate the migration of endothelial cells and protect adherent junctions. Cer is involved in programmed endothelial cell death, senescence and cell cycle arrest, leading to increased vascular permeability endothelial barrier dysfunction.

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
