# Peer review of "Unbalanced Sphingolipid Metabolism and Its Implications for the Pathogenesis of Psoriasis"

_molecules, 2020, doi:10.3390/molecules25051130_

Round 1
Reviewer 1 Report
General Comments
In the manuscript entitled ‘unbalanced sphingolipid metabolism and its implications for the pathogenesis of psoriasis’, the authors summarized the role of sphingolipids in the pathogenesis of skin psoriasis. Although the authors provide an extensive insight into the subject, addressing following minor concerns will be needed. Particularly, the authors need to add relevant references throughout the manuscript.
Minor issues:
In Figure 1 (line 73), provide numbering for each pathway. In line 238, additional citations should be needed to support the statement that barrier insults lead to excessive immune aberration towards Th23 since the provided citation only addresses Th17. In line 126-131 (Underlying the stratum~ and differentiation.), reference is needed. In line 166-169 (S1P inhibits~ re-epithelialization of wounds.), reference is needed. In line 188-191 (PsO, considered to~ world’s population.), reference is needed. In line 258-261 (Studies on programmed cell death~ on keratinocytes.), reference is needed. In line 261-263 (Many studies~ this process.), additional references are needed. In line 419-421 (Research suggests that~ mainly from the ER.), reference is needed. In section 3.2. Sphingolipid Action in Inflammation – Focus on the Molecular Mechanism of Psoriasis, text related to various immune cells is not divided by immune cell types. Thus, I would suggest dividing the text into more than two cell types: for instance, ①The crosstalk between SLs and DCs, ②The crosstalk between SLs and other immune cells (T cell, innate immune cells). In line 581-584 (S1P is crucial~ by release of VEGF and interleukins.), reference is needed. In line 584-585 (Beside VEGF~ TNFa and EGF.), reference is needed. In line 585-586 (S1P also activates~ cell migration.), reference is needed.Author Response
REVIEWER #1:
General Comments
In the manuscript entitled ‘unbalanced sphingolipid metabolism and its implications for the pathogenesis of psoriasis’, the authors summarized the role of sphingolipids in the pathogenesis of skin psoriasis. Although the authors provide an extensive insight into the subject, addressing following minor concerns will be needed. Particularly, the authors need to add relevant references throughout the manuscript.
Minor issues:
In Figure 1 (line 73), provide numbering for each pathway.
As demanded, it is now added (Figure 1 – line 75).
In line 238, additional citations should be needed to support the statement that barrier insults lead to excessive immune aberration towards Th23 since the provided citation only addresses Th17.
According to the Reviewer request, specific reference is now stated on page 7, line 268.
In line 126-131 (Underlying the stratum~ and differentiation.), reference is needed.
According to the Reviewer request, specific reference is now stated on page 4, line 135.
In line 166-169 (S1P inhibits~ re-epithelialization of wounds.), reference is needed.
According to the Reviewer request, specific reference is now stated on page 5, line 176.
In line 188-191 (PsO, considered to~ world’s population.), reference is needed.
According to the Reviewer request, specific reference is now stated on page 6, line 208.
In line 258-261 (Studies on programmed cell death~ on keratinocytes.), reference is needed. According to the Reviewer request, specific reference is now stated on page 7, line 291.
In line 261-263 (Many studies~ this process.), additional references are needed.
According to the Reviewer request, specific reference is now stated on page 7, line 293.
In line 419-421 (Research suggests that~ mainly from the ER.), reference is needed.
According to the Reviewer request, specific reference is now stated on page 11, line 479.
In section 3.2. Sphingolipid Action in Inflammation – Focus on the Molecular Mechanism of Psoriasis, text related to various immune cells is not divided by immune cell types. Thus, I would suggest dividing the text into more than two cell types: for instance, ①The crosstalk between SLs and DCs, ②The crosstalk between SLs and other immune cells (T cell, innate immune cells).
We thank the Reviewer for this suggestion, however, we decided to keep the text as it is. The participation of many various immune cells that interplay with each other and all contribute to the disease development, causing an amplified feedback loop, seems reasonable for not sharing the text as suggested by the Reviewer. Authors believe that directing potential readers in a way separately into areas referring to the cross-talk of the immunological factors, constantly communicating with each other by either establishing contacts or sending signals, may be tricky.
Reviewer 2 Report
Interesting review and really well written.
But this comes across as a narrative review failing to give a clear picture of where the field is heading. One could structure this by introducing early that traditional lipids do not capture the risk of psoriasis as well as psoriasis associated vascular disease. However, oxidized lipids do capture. So one needs o look into alternate lipid molecules one being SLs (ofc course their function are different)
Moreover, discuss the role of SLs and their mediators in not only skin lesions but may be endothelial dysfunction of the major vessels (aorta, coronaries, carotid) which is highly prevalent in PSO patients.
Conclude with a message as to how SLs characterization can help one inform treatment or prevention or even risk stratification.
The conclusion should address
Author Response
REVIEWER #2:
Interesting review and really well written.
But this comes across as a narrative review failing to give a clear picture of where the field is heading. One could structure this by introducing early that traditional lipids do not capture the risk of psoriasis as well as psoriasis associated vascular disease. However, oxidized lipids do capture. So one needs o look into alternate lipid molecules one being SLs (ofc course their function are different)
Lipids, especially those unsaturated, are relatively unstable organic substances that are easily oxidized. Lipid oxidation can actually lead to the formation of potentially harmful chemical compounds, such as aldehydes, ketones or epoxides which contribute to a number of dysfunctions (atherosclerosis, cancer). However, based on our knowledge, quantitative changes in traditional lipids can also contribute to the development of pathological conditions. Therefore, in this manuscript, we aimed to focus on describing only the sphingolipid class and their disturbances connected with immunological and skin barrier abnormalities, while pointing to the fact that on the one hand SLs perform multiple functions in the pathogenesis process in psoriasis, on the other hand, they play the role of compounds, those characteristics can be used as a complementary or alternative way or remedy in the choice of treatment, moreover, perhaps even in risk stratification.
Moreover, discuss the role of SLs and their mediators in not only skin lesions but may be endothelial dysfunction of the major vessels (aorta, coronaries, carotid) which is highly prevalent in PSO patients.
Please find it the Chapter no. 3.3. Sphingolipids In Endothelial Function – Vascular Endothelium Abnormalities in the Development of Psoriasis
Conclude with a message as to how SLs characterization can help one inform treatment or prevention or even risk stratification.
We agree with the Reviewer at this point. In addition to the fact that SLs perform multiple functions in the pathogenesis process in psoriasis, another level regarding the role of these compounds, may be the use of their characteristics as a complementary or alternative way or remedy in the choice of treatment. In this light, for example, study of the efficacy and safety of ponesimod (a selective modulator of the S1P receptor 1) suggested that sphingolipid metabolism may become a new target for the pharmacological treatment of psoriasis (Borodzicz et al., 2016). It is also known that proper treatment of psoriasis is the best strategy for preventing it from spreading. Perhaps in the future this approach will prove to be even helpful in assessing the risk of this disease.
Currently, this information is now added on page 16, lines 722 – 730.
Round 2
Reviewer 2 Report
Accept as is